# Multi Perspective Actor Critic: Adaptive Value Decomposition for Robust and Safe Reinforcement Learning

## Abstract

Real-world deployment of reinforcement learning requires simultaneously handling multiple objectives, safety constraints, and model uncertainty, yet existing methods address these challenges in isolation. We present Multi-Perspective Actor-Critic (MPAC), a novel framework that integrates all three aspects. MPAC combines value decomposition with component-specific risk assessment, enabling different objectives to maintain appropriate uncertainty tolerance, with collision avoidance employing extreme conservatism while efficiency permits optimistic planning. A novel influence-based mechanism dynamically adjusts component weights based on their decision relevance and learning progress, eliminating the need for fixed weights or prior reward knowledge. This yields policies that are simultaneously safe, robust to model perturbations, and less conservative than prior approaches. We prove that MPAC converges to a fixed point corresponding to a distributionally robust optimization problem with component-specific ambiguity sets, providing theoretical justification for its design. Empirically, across continuous-control benchmarks with safety constraints and perturbed dynamics, MPAC achieves superior Pareto trade-offs: it maintains high reward while matching or exceeding safety baselines. These results demonstrate that adaptively weighting decomposed objectives under uncertainty is a principled and practical path toward robust safe RL.

## 1 Introduction

Real-world reinforcement learning (RL) must simultaneously solve three challenges that are rarely addressed in combination. Agents must balance multiple competing objectives (Hayes et al., 2022), satisfy strict safety constraints (Kushwaha et al., 2025; Achiam et al., 2017), and remain robust to model uncertainty at deployment (Mankowitz et al., 2020; Tobin et al., 2017; Nilim & Ghaoui, 2005). Consider an autonomous vehicle navigating city streets. It must trade off travel time, energy consumption, and passenger comfort (multiple objectives), ensure collision avoidance (safety constraints), and operate reliably under changing road conditions, weather, and vehicle dynamics that differ from training simulations (model uncertainty). These challenges are deeply intertwined. Uncertainty affects objectives differently (minor energy estimation errors are tolerable, but safety violations are catastrophic), and the relative importance of objectives shifts with context (speed matters on highways, precision matters in parking). Addressing them separately leads to suboptimal or unsafe solutions.

Current approaches cover only parts of this puzzle. Multi-objective RL techniques such as value decomposition learn separate critics for different objectives, but combine them with fixed weights and without robustness, which leads to brittle performance under dynamics shift. Safe RL frameworks (Achiam et al., 2017; Ha et al., 2020) enforce cost constraints, but they treat performance through a single scalar reward and generally assume training and deployment dynamics match, leaving them brittle under model uncertainty. Domain randomization (Tobin et al., 2017) improves robustness by training on varied environments, but this requires large computational resources and still fails to guarantee safety. Risk-averse methods such as RAMU (Queeney & Benosman, 2023) handle model uncertainty and costs within CMDPs, but apply the same conservatism across all objectives, reducing performance on those that are tolerant to uncertainty. No existing paradigm can simultaneously

respect multiple objectives with different levels of robustness, adapt their relative importance across states, and guarantee safety under uncertainty.

We present *Multi-Perspective Actor-Critic* (MPAC), a framework designed to address these challenges in an integrated way. The key idea is that each objective should be evaluated under its own risk measure, matched to its uncertainty tolerance, and that their relative importance should adapt dynamically based on state and learning progress. MPAC maintains separate risk-aware critics for each objective, combines them with adaptive influence-based weights rather than fixed coefficients, and enforces safety through a dedicated conservative critic. This yields robust multi-objective performance with explicit safety guarantees, while training on a single environment without adversarial optimization or costly domain randomization.

Our contributions are threefold. First, we introduce a unified framework for multi-objective RL that integrates safety constraints and model uncertainty, enabling different objectives to adopt appropriate levels of conservatism. Second, we develop an adaptive weighting mechanism that balances objectives based on their current influence, removing the need for manual weight tuning or prior decomposition knowledge. Third, we provide theoretical analysis proving convergence to a fixed point and showing that MPAC implicitly solves a distributionally robust optimization problem with component-specific uncertainty sets. Extensive experiments demonstrate that MPAC consistently improves reward while maintaining strong safety guarantees, outperforming robust, safe, and multi-objective baselines across diverse continuous control tasks.

By allowing each objective to maintain its own perspective on risk while dynamically balancing their contributions, MPAC closes a gap between multi-objective RL, robust RL, and safe RL. This integration is essential for deploying reinforcement learning in safety-critical domains such as autonomous driving, robotics, and energy systems.

## 2 PRELIMINARIES

**Reinforcement Learning.** Reinforcement learning (RL) studies how agents can learn to act in sequential decision-making problems through interaction with an environment. Such problems are typically formalized as a Markov Decision Process (MDP) (Bellman, 1957), defined by a tuple $(\mathcal{S}, \mathcal{A}, P, R, \gamma)$, where $\mathcal{S}$ is the state space, $\mathcal{A}$ the action space, $P(\cdot \mid s, a)$ the transition dynamics, $R : \mathcal{S} \times \mathcal{A} \to \mathbb{R}$ the reward function, and $\gamma \in (0, 1)$ the discount factor. The goal is to learn a policy $\pi(a \mid s)$ that maximizes the expected discounted return

$$J(\pi) = \mathbb{E}_\pi \left[ \sum_{t=0}^\infty \gamma^t R(s_t, a_t) \right] \tag{1}$$

Central to many RL algorithms is the state–action value function (Q-function),

$$Q^\pi(s, a) = \mathbb{E}_\pi \left[ \sum_{t=0}^\infty \gamma^t R(s_t, a_t) \,\middle|\, s_0 = s, a_0 = a \right] \tag{2}$$

which measures the long-term value of executing action $a$ in state $s$ and following policy $\pi$ thereafter. Actor–Critic methods build directly on this formulation by combining a parameterized policy (the actor) with a value function estimator (the critic). The critic provides gradient information based on $Q^\pi$, enabling the actor to update in a direction that improves expected return while reducing variance compared to pure policy-gradient methods.

**Constrained MDPs and Safe Reinforcement Learning** Many real-world applications require policies that not only maximize performance but also satisfy safety constraints throughout deployment. This requirement is formalized through Constrained Markov Decision Processes (CMDPs) (Altman, 1999), which extend standard MDPs by introducing cost functions alongside rewards. A CMDP is defined as $(\mathcal{S}, \mathcal{A}, P, R, C, \gamma)$, where $C : \mathcal{S} \times \mathcal{A} \to \mathbb{R}$ represents costs associated with constraint violations, and other elements follow standard MDP notation.

The CMDP objective seeks to maximize expected return while maintaining costs below a threshold:

$$\max_\pi \mathbb{E}_\pi \left[ \sum_{t=0}^\infty \gamma^t R(s_t, a_t) \right] \quad \text{s.t.} \quad \mathbb{E}_\pi \left[ \sum_{t=0}^\infty \gamma^t C(s_t, a_t) \right] \leq B \tag{3}$$

where $B$ is the safety budget representing the maximum acceptable cumulative cost.

Recent safe RL algorithms address this challenge through Lagrangian relaxation (Stooke et al., 2020), trust region methods (Achiam et al., 2017), and constraint rectification (Xu et al., 2021). However, these methods assume that rewards can be represented by a single scalar, which obscures trade-offs between multiple objectives. They also train in a single environment, neglecting the model uncertainty faced during deployment. Finally, they treat objectives with fixed priorities and cannot adaptively rebalance performance and safety as the situation changes.

**Model Uncertainty and Risk Measures**  Classical RL and safe RL methods are built on the assumption that the transition dynamics observed during training are the same as those encountered at deployment. In practice this assumption often breaks down, since real-world environments can differ in ways that cannot be removed simply by collecting more data (Queeney & Benosman, 2023). As a result, policies that perform well during training may fail to maintain either high rewards or safety once deployed under shifted dynamics. Robust RL methods address this challenge by preparing policies for a range of possible environments. Domain randomization (Tobin et al., 2017) trains across perturbed dynamics, and adversarial methods optimize against worst-case perturbations. While effective in some cases, these approaches can be computationally expensive or overly conservative. A more principled alternative is to evaluate outcomes with *risk measures*, which generalize expectations to capture preferences toward risk or conservatism.

Formally, a risk measure $\rho : \mathcal{Z} \to \mathbb{R}$ maps a random variable $Z$ (such as uncertain returns) to a scalar, enabling pessimistic or optimistic evaluation depending on the choice of $\rho$. Coherent distortion risk measures (Artzner et al., 1999; Wang, 1996) satisfy four axioms that ensure consistent behavior and enable efficient computation through standard Bellman updates (see Appendix C for detailed properties).

Risk-Averse Model Uncertainty (RAMU) (Queeney & Benosman, 2023) applies these measures in safe RL by representing dynamics through uncertainty sets $\mathcal{P}_{s,a}$ over transitions for each state–action pair. For a policy $\pi$ and risk measures $\rho^+$ (for rewards) and $\rho^-$ (for costs), the Bellman operators become

$$\mathcal{T}^\pi_{\rho^+, R} Q(s,a) := R(s,a) + \gamma \rho^+_{p \sim \mathcal{P}_{s,a}} \left[ \mathbb{E}_{s' \sim p} \mathbb{E}_{a' \sim \pi(\cdot|s')} [Q(s',a')] \right] \tag{4}$$

$$\mathcal{T}^\pi_{\rho^-, C} Q(s,a) := C(s,a) + \gamma \rho^-_{p \sim \mathcal{P}_{s,a}} \left[ \mathbb{E}_{s' \sim p} \mathbb{E}_{a' \sim \pi(\cdot|s')} [Q(s',a')] \right] \tag{5}$$

where $\mathcal{P}_{s,a}$ captures the distribution over plausible transition models. RAMU shows that this formulation is equivalent to solving a distributionally robust optimization problem, while remaining tractable through sample-based approximation.

**Value Decomposition in Reinforcement Learning**  Many real-world tasks involve multiple, often conflicting objectives that cannot be captured by a single scalar reward. For instance, an autonomous driving agent must trade off speed, passenger comfort, energy efficiency, and safety, each of which may carry different priorities and risk tolerances. To capture such problems, multi-objective reinforcement learning (MORL) represents rewards as vectors $\mathbf{R} = (R_1, \ldots, R_m)$ rather than a single scalar (Roijers et al., 2013).

Value decomposition methods (Russell & Zimdars, 2003; MacGlashan et al., 2022) provide a practical way to learn from vector-valued rewards while maintaining a single deployable policy. For each component $R_i$, they define a component-specific value function

$$Q^\pi_i(s,a) = \mathbb{E}_{\pi, P} \left[ \sum_{t=0}^\infty \gamma^t R_i(s_t, a_t) \,\middle|\, s_0 = s, a_0 = a \right], \tag{6}$$

and combine them into a composite value function $Q^\pi(s,a) = \sum_{i=1}^m w_i Q^\pi_i(s,a)$, with weights $w_i \geq 0$ reflecting the relative importance of each objective.

This decomposition is crucial in the context of robustness. By separating objectives into distinct value functions, it enables the use of different risk measures for different components. Classical RL and robust CMDP methods such as RAMU apply the same robustness treatment across all objectives, whereas decomposition makes it possible to tailor robustness to the specific requirements of each objective—a key capability that MPAC builds upon.

## 3 PROBLEM STATEMENT

Our goal is to learn a policy that maximizes expected return while remaining safe under model uncertainty. We formalize this with a *Constrained Multi-Objective MDP* (C-MOMDP)

$$M = (\mathcal{S}, \mathcal{A}, P, R, C, \gamma),$$

where $R : \mathcal{S} \times \mathcal{A} \to \mathbb{R}^m$ is a vector-valued reward with components $\{R_i\}_{i=1}^m$ representing distinct objectives, and $C : \mathcal{S} \times \mathcal{A} \to \mathbb{R}$ encodes costs associated with safety violations. The objective is to maximize the expected aggregated return derived from $R$ while ensuring the expected cumulative cost does not exceed a budget $B$.

To capture deployment-time uncertainty, we extend the C-MOMDP with a distribution $\mathcal{P}$ over transition models. For each state–action pair $(s, a)$, the set $\mathcal{P}_{s,a}$ defines a distribution over possible transition probability functions, capturing how dynamics may differ between training and deployment. We assume that $\mathcal{P}$ is *rectangular*, meaning that uncertainty at each state–action pair is independent (Nilim & El Ghaoui, 2005; Xu & Mannor, 2010). This assumption permits decomposition of the optimization problem across state–action pairs while retaining meaningful robustness guarantees. Intuitively, this means that the variability in transition models depends only on the current state–action pair, not on others. While this independence assumption is common and often practical in real-world domains, it can be violated in adversarial or highly coupled settings.

## 4 MULTI-PERSPECTIVE ACTOR CRITIC

We present *Multi-Perspective Actor-Critic (MPAC)*, a novel framework for multi-objective reinforcement learning under model uncertainty. Our approach builds on three principles: (1) different objectives require different levels of risk awareness when facing uncertainty, (2) the relative importance of objectives should adapt based on both the current state and the policy's evolving capabilities rather than remain fixed, and (3) safety constraints must be maintained as hard constraints separate from reward optimization. These principles lead to a unified framework that addresses the limitations of existing approaches while maintaining computational efficiency.

### 4.1 COMPONENT-SPECIFIC RISK-AWARE CRITICS

Existing approaches capture only parts of the deployment challenge. Robust RL methods such as RAMU (Queeney & Benosman, 2023) address model uncertainty through risk-averse value estimation, but they assume a *single scalar reward*. This prevents applying different levels of risk tolerance to different objectives. Conversely, multi-objective RL (MORL) methods (MacGlashan et al., 2022) decompose vector-valued rewards into separate critics, improving interpretability and credit assignment, but they neither handle model uncertainty nor enforce safety constraints. As a result, they implicitly assume identical risk tolerance across objectives, ignoring that safety-critical components often demand far stronger guarantees than performance objectives.

MPAC unifies these perspectives. By combining multi-objective decomposition with risk-aware estimation, it enables each reward component to be evaluated under its own risk profile: safety-critical objectives adopt highly conservative measures, while performance-oriented objectives retain more optimistic evaluation. Formally, for each reward component $R_i$, we maintain a critic

$$Q_i^\pi(s, a) = R_i(s, a) + \gamma \, \rho_i^+ \big[ \mathbb{E}_{s' \sim p_{s,a}} \mathbb{E}_{a' \sim \pi(\cdot|s')} \big[ Q_i^\pi(s', a') \big] \big]_{p_{s,a} \sim \mathcal{P}_{s,a}} \tag{7}$$

where $\rho_i^+$ is a coherent distortion risk measure chosen for component $i$ according to its uncertainty tolerance, and $\mathcal{P}_{s,a}$ is the distribution over possible transition dynamics.

Safety is represented through a dedicated cost critic with a strongly risk-averse measure $\rho^-$:

$$Q_C^\pi(s, a) = C(s, a) + \gamma \, \rho^- \big[ \mathbb{E}_{s' \sim p_{s,a}} \mathbb{E}_{a' \sim \pi(\cdot|s')} \big[ Q_C^\pi(s', a') \big] \big]_{p_{s,a} \sim \mathcal{P}_{s,a}} \tag{8}$$

This critic structure allows heterogeneous objectives and hard safety constraints to coexist, something neither robust RL nor MORL alone can achieve. It forms the foundation for the adaptive weighting mechanism introduced in the next subsection.

## 4.2 DYNAMIC INFLUENCE-BASED WEIGHTING

Traditional value decomposition methods (MacGlashan et al., 2022; Ding et al., 2024) combine component Q-functions using fixed weights: $Q_{\text{total}}(s, a) = \sum_i w_i Q_i(s, a)$. These fixed weights $w_i$ are typically assumed to be known constants that reflect the original reward decomposition. In practice, however, the underlying decomposition is rarely observable—most environments expose only a scalar reward signal, not its constituent objectives (Hayes et al., 2022). Even if components are manually specified, fixed weights assume that the relative importance of objectives is constant across all states and throughout training. This creates two problems: (i) state-dependent priorities (e.g., speed is critical on highways but precision matters when parking) cannot be represented, and (ii) during training, unreliable critics may dominate decisions before they have learned meaningful structure. Together, these limitations motivate an adaptive scheme that can respond to context and learning progress.

MPAC addresses this challenge with *influence-based dynamic weighting*, where each component's weight reflects how strongly it shapes action selection in the current state. The key intuition is that a critic that assigns markedly different values to available actions is signaling strong guidance. It has identified distinctions that matter for decision-making. Conversely, a critic whose values are nearly identical across actions conveys little information and should be down-weighted. By measuring this degree of action differentiation, we allow components to earn influence when they provide useful guidance, while those that are noisy or indifferent naturally contribute less. This principle aligns with the broader idea that the most informative signals for policy improvement are those that discriminate between competing actions.

We formalize this intuition through a gradient-based influence metric (MacGlashan et al., 2022):

$$I_i^\pi(s; \theta) = \|\nabla_a Q^\pi(s, a; \theta) - \nabla_a Q_{\neg i}^\pi(s, a; \theta)\|_2 \Big|_{a=\pi(s)} \tag{9}$$

where $Q_{\neg i}^\pi = \sum_{j \neq i} \phi_j^\pi(s) Q_j^\pi$ represents the composite Q-function excluding component $i$. This measures how removing component $i$ changes the action gradient—high influence indicates the component critically affects action selection. Components with steep value landscapes that differ from the aggregate naturally receive higher influence.

Influences are converted to weights through softmax normalization:

$$\phi_i^\pi(s) = \frac{\exp(I_i^\pi(s)/\tau)}{\sum_{j=1}^m \exp(I_j^\pi(s)/\tau)} \tag{10}$$

with temperature parameter $\tau > 0$ controlling sharpness. This guarantees positive, normalized weights while retaining smooth differentiability and prevents any component from being completely ignored (avoiding potential loss of information).

At first glance, this scheme introduces a circular dependency: weights depend on Q-functions, which themselves depend on the weights. In fact, this is a desirable property. The alternating updates between critics and weights define a fixed-point iteration where components gradually earn influence by demonstrating consistent value under their current weighting. We prove in Section 5 (Appendix B) that under mild conditions this process converges to a stable configuration, analogous to other fixed-point methods such as value iteration or attention mechanisms.

This design also yields useful emergent behavior. Early in training, influences are similar and weights remain near-uniform, preventing unreliable critics from dominating. As training progresses, components that capture reliable structure automatically gain influence, producing a self-organizing curriculum without manual scheduling.

## 4.3 MPAC OBJECTIVE AND TRAINING ALGORITHM

The component-specific critics (Section 4.1) and adaptive weights (Section 4.2) combine into a composite Q-function:

$$Q_{\text{composite}}^\pi(s, a) = \sum_{i=1}^m \phi_i^\pi(s) Q_i^\pi(s, a), \tag{11}$$

---

**Algorithm 1** Multi-Perspective Actor-Critic (MPAC)

---

**Require:** Replay buffer $\mathcal{D}$, risk measures $\{\rho_i^+\}_{i=1}^m$, cost risk measure $\rho^-$, safety budget $B$
1: Initialize policy $\pi$, component critics $\{Q_i\}_{i=1}^m$, cost critic $Q_c$, and weights $\phi_i^\pi(s)$
2: **for** each training iteration **do**
3:     Sample batch $(s, a, \mathbf{r}, c, s')$ from $\mathcal{D}$
4:     Generate perturbed next states $\tilde{s}' \sim \mathcal{P}_{s,a}$ to model uncertainty
5:     Update each component critic $Q_i$ with risk-adjusted targets using $\rho_i^+$
6:     Update cost critic $Q_c$ with strongly risk-averse targets using $\rho^-$
7:     Compute influence scores $I_i^\pi(s)$ (Eq. 9) and update weights $\phi_i^\pi(s)$ (Eq. 10)
8:     Update policy $\pi$ by maximizing $Q_{\text{composite}}^\pi(s, a)$ subject to the safety constraint $J_{\rho^-,C}(\pi) \leq B$
        (Eq. 14)
9: **end for**
10: **return** Final policy $\pi$ and critics $\{Q_i\}_{i=1}^m, Q_c$

---

with corresponding value function

$$V_{\text{MPAC}}(\pi) = \mathbb{E}_{s \sim d^\pi}\left[\sum_{i=1}^m \phi_i^\pi(s) V_i^\pi(s)\right], \tag{12}$$

where $d^\pi$ is the stationary state distribution and $V_i^\pi(s) = \mathbb{E}_{a \sim \pi(\cdot|s)}[Q_i^\pi(s, a)]$.

Although the weights $\phi_i^\pi(s)$ depend on the policy, this objective is *well-defined*. For any fixed policy $\pi$, the influence computation produces deterministic weights, yielding a scalar $V_{\text{MPAC}}(\pi) \in \mathbb{R}$. The alternating updates of critics and weights define a fixed-point iteration, whose convergence is analyzed in Section 5 (Appendix B). This ensures that MPAC optimizes a coherent and stable objective rather than an ill-posed one.

The full optimization problem is

$$\boxed{\begin{aligned} \pi^* = \arg\max_{\pi \in \Pi} \quad & V_{\text{MPAC}}(\pi) \\ \text{s.t.} \quad & J_{\rho^-,C}(\pi) \leq B, \end{aligned}} \tag{13}$$

where $J_{\rho^-,C}(\pi) = \mathbb{E}_{s_0 \sim d_0}[Q_C^\pi(s_0, \pi(s_0))]$ is the risk-averse expected cost and $B$ is the safety budget from Eq. 3.

In practice, we solve this problem off-policy through alternating updates of critics, weights, and policy. Risk-aware critics are trained using coherent distortion risk measures, which reweight Bellman backups according to a distortion function $g$ (Queeney & Benosman, 2023). To capture deployment-time variability, we use $\mathcal{P}_{s,a}$, a distribution over perturbed next states that models uncertainty in dynamics. This requires only generative access to a single training environment, avoids adversarial setups, and supports domain-specific perturbations when available. The policy improvement step is then carried out as

$$\begin{aligned} \max_\pi \quad & \mathbb{E}_{s \sim \mathcal{D}}\left[\mathbb{E}_{a \sim \pi(\cdot|s)}[Q_{\text{composite}}^\pi(s, a)]\right] \\ \text{s.t.} \quad & \mathbb{E}_{s \sim \mathcal{D}}\left[\mathbb{E}_{a \sim \pi(\cdot|s)}[Q_c^\pi(s, a)]\right] \leq B, \end{aligned} \tag{14}$$

where $\mathcal{D}$ denotes the replay buffer. Full mathematical and implementation details are provided in Appendix D.

Algorithm 1 summarizes the MPAC training loop. At each iteration, transitions are sampled from the replay buffer, and $\mathcal{P}_{s,a}$ generates perturbed next states to model uncertainty. Component critics are updated using risk-adjusted targets derived from their associated distortion measures, while the cost critic is updated with a strongly risk-averse measure to enforce constraints. The gradient-based influence metric is then computed and converted into adaptive weights via softmax, after which the policy is improved by solving Eq. 14. This structure mirrors standard safe actor-critic methods, with MPAC's modifications confined to the critic decomposition and dynamic weight adaptation.

## 5 THEORETICAL ANALYSIS

In this section, we establish the theoretical foundations of MPAC. We first show that despite the dynamic, policy-dependent nature of the influence-based weights, the MPAC iteration admits fixed points and converges to one under mild regularity assumptions. We then demonstrate that such fixed-point solutions can be interpreted as solving a distributionally robust optimization problem where each objective faces uncertainty matched to its own risk tolerance. Together, these results formalize MPAC's ability to achieve component-specific robustness while maintaining computational efficiency.

**Theorem 1** (Convergence of MPAC). *Assume compact state and action spaces, bounded rewards and costs, stochastic policies drawn from a convex and compact policy class, continuous influence functions, and uniqueness of the constrained policy optimization solution for each fixed set of weights. Then the MPAC update operator admits at least one fixed point $(\pi^*, \phi^*)$. Moreover, the alternating update procedure converges to such a fixed point under these assumptions.*

The proof applies the Kakutani–Fan–Glicksberg fixed-point theorem (Kakutani, 1941; Glicksberg, 1952) to the MPAC update operator, showing that the composition of influence computation and constrained policy optimization defines a closed mapping on a compact convex set (see Appendix A for the complete argument). This guarantees the existence of at least one fixed point. Convergence of the alternating updates to such a fixed point requires the additional uniqueness assumption, which ensures that the policy and influence-based weights become mutually consistent.

Having established convergence, we now characterize what MPAC converges to. The fixed-point solution admits a natural interpretation in terms of distributionally robust optimization:

**Theorem 2** (Distributionally Robust Characterization). *Let $(\pi^*, \phi^*)$ be a fixed point of the MPAC iteration under rectangular uncertainty. Then $\pi^*$ is the optimal solution to*

$$\max_{\pi} \sum_{i=1}^{m} \mathbb{E}_{s \sim d^\pi}\left[\phi_i^*(s) \min_{P_i \in \mathcal{U}_i} \mathbb{E}_{P_i}[V_i^\pi(s)]\right] \quad s.t. \quad \max_{P_c \in \mathcal{U}_c} \mathbb{E}_{P_c}[V_c^\pi(s)] \leq B, \tag{15}$$

*where $\mathcal{U}_i$ and $\mathcal{U}_c$ are the ambiguity sets induced by risk measures $\rho_i^+$ and $\rho^-$, respectively.*

This result shows that MPAC implicitly solves a distributionally robust problem in which each objective accounts for its own worst-case environment, a key advantage over approaches that impose the same robustness level on all objectives. The proof relies on the duality between coherent risk measures and ambiguity sets over probability distributions (Iyengar, 2005; Nilim & El Ghaoui, 2005; Shapiro et al., 2021), demonstrating that the risk-aware Bellman equations correspond to robust Bellman equations with component-specific uncertainty sets (see Appendix B).

These theoretical results provide a solid foundation for MPAC's practical effectiveness. The convergence guarantee ensures reliable training despite the circular dependence between weights and values. The distributionally robust characterization explains why MPAC achieves superior empirical performance, since it tailors robustness to each objective's needs rather than applying a uniform penalty. Crucially, this is accomplished without explicit minimax optimization or domain randomization, preserving computational efficiency while providing formal guarantees.

## 6 EXPERIMENTS

We evaluate MPAC on five continuous control tasks from the Real-World RL Suite (Dulac-Arnold et al., 2020): Cartpole Swingup, Walker Walk/Run, and Quadruped Walk/Run. These environments provide naturally multi-objective rewards together with safety costs, making them ideal for studying the intersection of robustness, safety, and multi-objective optimization. All algorithms are trained for 500k steps on the *nominal environment only*, then evaluated under perturbed dynamics where key physical parameters (e.g., mass, friction, actuator strength) vary systematically from training values (see Appendix E). For each perturbation level, we run multiple seeds and aggregate results across environments, yielding statistically meaningful comparisons. Domain Randomization (Tobin et al., 2017) is the only method trained on perturbed environments, reflecting its standard setup, giving it more diverse training data than other baselines. Evaluation is always performed under perturbations (including the nominal case) to reflect inevitable model mismatches in real deployment.

Table 1: Performance aggregated across all environments under perturbed evaluation. Safety rate is the percentage of episodes satisfying the cost budget (cost $\leq$ 100). Rewards and costs are normalized to the MPO baseline. MPAC achieves the best overall trade-off, with the highest reward and strong safety under perturbations.

| Algorithm | Safety Rate (%) | Reward (norm) | Cost (norm) |
|---|---|---|---|
| MPO | 6.9 | 1.00 | 1.00 |
| CRPO | 52.1 | 0.88 | 0.83 |
| Domain Rand | 7.8 | 1.10 | 1.02 |
| VD-Safe | **71.3** | 0.96 | **0.18** |
| RAMU | 26.7 | 1.10 | 0.64 |
| **MPAC** | 62.3 | **1.16** | 0.23 |

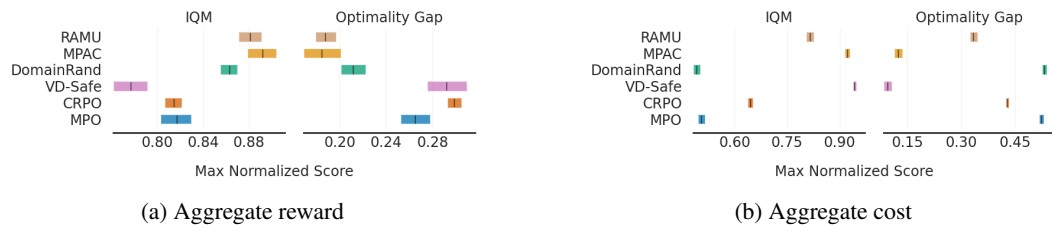

(a) Aggregate reward  (b) Aggregate cost

Figure 1: Interquartile mean (IQM) scores with 95% confidence intervals across tasks and perturbations. Higher reward and lower cost are better. MPAC achieves the best reward IQM while maintaining costs comparable to VD-Safe, with narrow intervals confirming consistency across seeds and perturbation levels.

Following Agarwal et al. (2021), we use the `rliable` library for statistical analysis, employing interquartile mean (IQM) scores and performance profiles to ensure robust conclusions.

We compare against several representative baselines: MPO (Abdolmaleki et al., 2018) as the unconstrained RL baseline, CRPO (Xu et al., 2021) as a constrained RL method, Domain Randomization (Tobin et al., 2017) as a traditional robustness approach, VD-Safe combining value decomposition with CRPO for multi-objective safe RL, and RAMU (Queeney & Benosman, 2023) for robust and safe single-objective RL. All methods use identical architectures and training procedures (Appendix D).

Table 1 and Figure 1 together illustrate the trade-offs across methods. All values are normalized to MPO, which serves as the reference point. MPO itself ignores both safety and robustness, leading to extremely low safety satisfaction (6.9%) and providing no guarantee under perturbations. CRPO explicitly enforces constraints, improving safety to 52.1%, but it pays with reduced reward (0.88), particularly because it has no mechanism for robustness. Domain Randomization addresses robustness by training across perturbed environments, yielding higher reward (1.10) but almost no safety (7.8%), while requiring significantly more computation. VD-Safe, which augments CRPO with value decomposition, achieves the best safety rate (71.3%) and improves reward compared to CRPO (0.96 vs. 0.88), showing the benefit of value decomposition; yet its fixed weights and lack of robustness lead to overly conservative policies. RAMU introduces risk-aware critics to handle uncertainty and costs jointly, improving both metrics relative to CRPO, but applies uniform conservatism to all objectives, limiting safety (26.7%) and reward potential. MPAC achieves the most favorable balance, with 62.3% safety that remains robust across perturbations and only nine points lower than the most conservative method, while at the same time reaching the highest reward (1.16), a 21% gain over MPO. By assigning different levels of risk sensitivity to different objectives and adjusting their weights dynamically, MPAC avoids the over-conservatism of VD-Safe and the uniform treatment in RAMU, resulting in robust multi-objective performance. The IQM curves in Figure 1 confirm that these improvements hold consistently across seeds and perturbation levels, with MPAC's reward distribution clearly separated from the baselines while keeping costs comparable to VD-Safe.

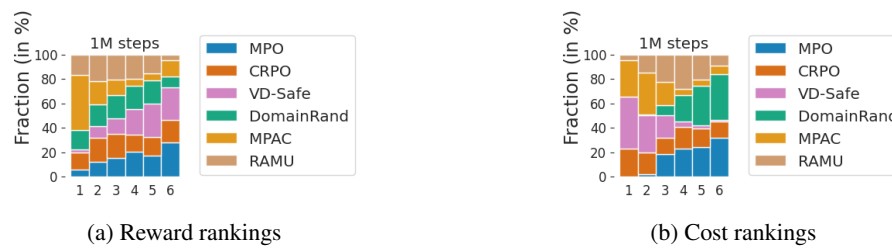

(a) Reward rankings                 (b) Cost rankings

Figure 2: Ranking distributions across all environments and perturbations. Lower rank is better. MPAC consistently achieves top reward ranks while maintaining strong cost ranks, confirming robustness across settings.

The ranking analysis in Figure 2 further emphasizes MPAC's reliability. Rather than excelling in a single metric or environment, MPAC consistently appears near the top across both reward and cost rankings. It secures almost 40% of first-place positions in reward and over 60% of top-two placements, while also ranking within the top three for cost in most settings. Other methods tend to specialize, with MPO and Domain Randomization occasionally achieving high reward ranks but failing on safety, and VD-Safe and CRPO doing the opposite. RAMU achieves a better balance but remains less consistent than MPAC. These distributions show that MPAC does not win by averaging across outliers but by providing strong performance across environments and perturbations.

Additional analyses in Appendix F reinforce these findings. Dynamic-vs-fixed comparisons confirm the benefits of adaptive weighting, robustness curves demonstrate that MPAC maintains performance under increasing perturbations, and ranking distributions across environments show that MPAC consistently achieves top reward ranks while preserving strong safety guarantees.

## 7 CONCLUSION

We introduced Multi-Perspective Actor-Critic (MPAC), a framework that unifies multi-objective reinforcement learning, safety constraints, and robustness to model uncertainty. MPAC rests on the insight that different objectives demand different levels of conservatism. Safety-critical components benefit from risk-averse treatment, while performance objectives can tolerate more optimistic planning. Our theoretical analysis established both the convergence of the MPAC iteration and its equivalence to a distributionally robust optimization problem with component-specific ambiguity sets, providing strong foundations for the approach.

Empirically, MPAC achieves superior trade-offs between performance and safety across diverse perturbations, consistently outperforming RAMU, CRPO, fixed-weight value decomposition, and domain randomization baselines. By adapting influence-based weights dynamically, MPAC avoids the limitations of fixed weighting schemes and leverages component signals when they are most reliable. While the benefits are most pronounced in complex, multi-faceted tasks, this matches MPAC's design for real-world scenarios where objectives and risks interact in non-trivial ways. Future work includes exploring richer non-linear decompositions and testing MPAC in real-world applications. By tailoring robustness per objective while ensuring constraint satisfaction, MPAC represents a step toward reinforcement learning that is both practically deployable and theoretically grounded.

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

## A  CONVERGENCE ANALYSIS

In this appendix we provide a detailed analysis of the MPAC iteration. We first formalize the update operator, then prove existence of fixed points and convergence of the iterative procedure under mild assumptions.

### A.1  FIXED-POINT FORMULATION

The MPAC algorithm alternates between (i) computing influence-based weights for a given policy, and (ii) updating the policy to maximize the weighted objective subject to safety constraints. We formalize this as a set-valued operator on the joint space of policies and weight functions.

**Definition 1** (MPAC Update Operator). Define $T : \Pi \times \Phi \to 2^{\Pi \times \Phi}$ by

$$T(\pi, \phi) = \Big\{ (\pi', \phi') : \phi_i'(s) = \frac{\exp(I_i^\pi(s)/\tau)}{\sum_{j=1}^m \exp(I_j^\pi(s)/\tau)},$$

$$\pi' \in \arg\max_{\tilde{\pi} \in \Pi} \mathbb{E}_{s \sim d^\pi} \Big[ \sum_{i=1}^m \phi_i(s) Q_i^{\tilde{\pi}}(s, \tilde{\pi}(s)) \Big] \tag{16}$$

$$\text{s.t. } \mathbb{E}_{s \sim d^\pi} [Q_c^{\tilde{\pi}}(s, \tilde{\pi}(s))] \le B \Big\}.$$

A fixed point of $T$ is a pair $(\pi^*, \phi^*)$ with $(\pi^*, \phi^*) \in T(\pi^*, \phi^*)$, corresponding exactly to the self-consistent solution described in the main text. The MPAC iteration is the sequence $(\pi^{(t)}, \phi^{(t)})$ defined by

$$(\pi^{(t+1)}, \phi^{(t+1)}) \in T(\pi^{(t)}, \phi^{(t)}). \tag{17}$$

## A.2 EXISTENCE OF FIXED POINTS

**Lemma 1** (Existence of MPAC Fixed Point). *Assume (i) state and action spaces are compact, (ii) rewards and costs are bounded, (iii) the policy class $\Pi$ is convex and compact, and (iv) the influence function is continuous. Then the MPAC iteration has at least one fixed point $(\pi^*, \phi^*)$.*

*Proof.* The operator $T$ has non-empty values (the constrained optimization admits a solution and the weight update is uniquely defined), convex values (the set of optimal stochastic policies is convex since expectations are linear in $\pi$), and is upper hemicontinuous (by continuity of influences and compactness of $\Pi$). Since $\Pi \times \Phi$ is compact and convex, the Kakutani–Fan–Glicksberg theorem guarantees existence of a fixed point. $\square$

## A.3 CONVERGENCE OF THE ITERATIVE PROCEDURE

**Theorem 3** (Convergence of MPAC Iteration). *Under the assumptions of Lemma 1, and further assuming that the constrained policy update has a unique optimizer and that the feasible set $\{\pi \in \Pi : \mathbb{E}[Q_c^\pi] \le B\}$ has non-empty interior, the sequence $(\pi^{(t)}, \phi^{(t)})$ generated by MPAC converges to a fixed point $(\pi^*, \phi^*)$ in the metric*

$$d((\pi_1, \phi_1), (\pi_2, \phi_2)) = \sup_{s \in \mathcal{S}} \|\pi_1(\cdot \mid s) - \pi_2(\cdot \mid s)\|_{TV} + \|\phi_1 - \phi_2\|_\infty,$$

*where $\|\cdot\|_{TV}$ denotes the total variation distance between probability distributions.*

*Sketch.* Define the potential function $\Phi(\pi, \phi) = \mathbb{E}_{s \sim d^\pi} \big[ \sum_i \phi_i(s) Q_i^\pi(s, \pi(s)) \big]$. Each policy update maximizes $\Phi$ given $\phi$, so the sequence of potential values is non-decreasing and bounded by $R_{\max}/(1 - \gamma)$. By the monotone convergence theorem, increments vanish asymptotically. Uniqueness of the policy optimizer implies $\sup_s \|\pi^{(t+1)}(\cdot \mid s) - \pi^{(t)}(\cdot \mid s)\|_{TV} \to 0$. Continuity of influence functions and Lipschitz continuity of the softmax mapping then imply $\|\phi^{(t+1)} - \phi^{(t)}\|_\infty \to 0$. Thus the sequence forms a Cauchy sequence in the product metric and converges to a fixed point. $\square$

# B DISTRIBUTIONALLY ROBUST CHARACTERIZATION

This appendix provides the proof of Theorem 2, showing that the fixed-point solution of MPAC corresponds to the optimal policy of a distributionally robust optimization problem with component-specific ambiguity sets.

## B.1 RISK MEASURES AND AMBIGUITY SETS

A well-known duality links coherent distortion risk measures and distributionally robust optimization (Shapiro et al., 2021; Ruszczyński & Shapiro, 2006). For any coherent risk measure $\rho$ with

distortion function $g$, there exists an ambiguity set $\mathcal{U}$ of probability distributions such that for any random variable $X$,

$$\rho[X] = \min_{Q \in \mathcal{U}} \mathbb{E}_Q[X]. \tag{18}$$

Intuitively, pessimistic distortion functions induce ambiguity sets that overweight unfavorable outcomes, while optimistic ones induce more uniform sets. In our setting, $\mathcal{P}_{s,a}$ denotes the uncertainty set of transition models, and applying $\rho$ corresponds to optimizing against the worst-case distribution in the induced ambiguity set.

### B.2 FIXED-POINT PROPERTIES OF MPAC

Let $(\pi^*, \phi^*)$ denote a fixed point of the MPAC iteration. At this point three conditions hold simultaneously:

$$\pi^* \in \arg\max_{\pi} \left\{ \mathbb{E}_{s \sim d^{\pi^*}} \left[ \sum_{i=1}^{m} \phi_i^*(s) Q_i^{\pi}(s, \pi(s)) \right] : \mathbb{E}_{s \sim d^{\pi^*}} [Q_c^{\pi}(s, \pi(s))] \leq B \right\}, \tag{19}$$

$$\phi_i^*(s) = \frac{\exp(I_i^{\pi^*}(s)/\tau)}{\sum_{j=1}^{m} \exp(I_j^{\pi^*}(s)/\tau)}, \tag{20}$$

$$Q_i^{\pi^*}(s, a) = r_i(s, a) + \gamma \rho_i^+ \left[ Q_i^{\pi^*}(s', \pi^*(s')) \right]_{P \sim \mathcal{P}_{s,a}}. \tag{21}$$

These express, respectively, that the policy optimizes the weighted risk-aware objective, the weights are consistent with influence, and each critic satisfies its risk-aware Bellman equation.

### B.3 PROOF OF THEOREM 2

By the dual representation in Eq. 18, each risk-aware Bellman equation (Eq. 21) can be rewritten as

$$Q_i^{\pi^*}(s, a) = r_i(s, a) + \gamma \min_{P_i \in \mathcal{U}_i} \mathbb{E}_{s' \sim P_i}[Q_i^{\pi^*}(s', \pi^*(s'))], \tag{22}$$

where $\mathcal{U}_i$ is the ambiguity set corresponding to risk measure $\rho_i^+$. Thus each $Q_i^{\pi^*}$ is the fixed point of a robust Bellman operator with ambiguity set $\mathcal{U}_i$ (Iyengar, 2005; Nilim & El Ghaoui, 2005). Equivalently,

$$V_i^{\pi^*}(s) = \min_{P_i \in \mathcal{U}_i} \mathbb{E}_{P_i} \left[ \sum_{t=0}^{\infty} \gamma^t r_i(s_t, \pi^*(s_t)) \,\big|\, s_0 = s \right]. \tag{23}$$

Combining with equation 19 and equation 20, the fixed-point policy $\pi^*$ maximizes

$$\sum_{i=1}^{m} \mathbb{E}_{s \sim d^{\pi^*}} \left[ \phi_i^*(s) \min_{P_i \in \mathcal{U}_i} \mathbb{E}_{P_i}[V_i^{\pi}(s)] \right], \tag{24}$$

subject to the safety constraint

$$\mathbb{E}_{s \sim d^{\pi^*}}[Q_c^{\pi^*}(s, \pi^*(s))] = \max_{P_c \in \mathcal{U}_c} \mathbb{E}_{P_c}[V_c^{\pi^*}(s)] \leq B, \tag{25}$$

where $\mathcal{U}_c$ is the ambiguity set induced by the conservative cost risk measure $\rho^-$. This establishes that $\pi^*$ solves the distributionally robust optimization problem claimed in Theorem 2.

## C COHERENT DISTORTION RISK MEASURES

Risk assessment is central to robust reinforcement learning, providing a principled way to evaluate outcomes under uncertainty. A risk measure $\rho : Z \to \mathbb{R}$ maps a random variable to a real number, summarizing its risk in a single value.

**Definition 2** (Coherent Risk Measure). A risk measure $\rho$ is coherent if it satisfies monotonicity, translation invariance, positive homogeneity, and convexity.

Coherent risk measures satisfy several key properties as characterized by Majumdar & Pavone (2020):

- Monotonicity: If $Z_1 \leq Z_2$ almost everywhere, then $\rho(Z_1) \leq \rho(Z_2)$
- Translation invariance: For any $\alpha \in \mathbb{R}$, $\rho(Z + \alpha) = \rho(Z) + \alpha$
- Positive homogeneity: For any $\lambda \geq 0$, $\rho(\lambda Z) = \lambda \rho(Z)$
- Convexity: $\rho(\lambda Z_1 + (1 - \lambda)Z_2) \leq \lambda \rho(Z_1) + (1 - \lambda)\rho(Z_2)$ for $\lambda \in [0, 1]$
- Comonotonic additivity: For comonotonic random variables $Z_1, Z_2$, $\rho(Z_1 + Z_2) = \rho(Z_1) + \rho(Z_2)$

**Definition 3** (Distortion Risk Measure). Let $g : [0, 1] \to [0, 1]$ be a non-decreasing, left-continuous function with $g(0) = 0$ and $g(1) = 1$. The distortion risk measure with distortion function $g$ is

$$\rho_g(Z) = \int_0^1 F_Z^{-1}(u) \, d\tilde{g}(u), \tag{26}$$

where $F_Z^{-1}$ is the quantile function of $Z$ and $\tilde{g}(u) = 1 - g(1 - u)$. The risk measure $\rho_g$ is coherent if and only if $g$ is concave.

Common examples are the expectation $g(u) = u$ (risk-neutral evaluation) and Conditional Value-at-Risk (CVaR) at level $\alpha \in (0, 1]$, with $g(u) = \min(u/\alpha, 1)$, which averages the worst $\alpha$ outcomes.

These measures admit a dual interpretation: each coherent distortion risk measure corresponds to an ambiguity set of probability distributions (Shapiro et al., 2021; Ruszczyński & Shapiro, 2006). Thus, applying $\rho_g$ to returns is equivalent to evaluating them under the worst-case distribution in the associated set, which underpins the distributionally robust view of MPAC.

In practice, risk-aware value estimation is implemented with finite samples. For $n$ samples sorted as $y_{(1)} \leq \ldots \leq y_{(n)}$, the empirical distortion risk is

$$\hat{\rho}_g = \sum_{i=1}^{n} w_i y_{(i)}, \quad w_i = g\left(\tfrac{i}{n}\right) - g\left(\tfrac{i-1}{n}\right). \tag{27}$$

This discretization enables efficient computation without requiring explicit ambiguity sets or adversarial optimization. By choosing different distortion functions $g$ for different reward components, MPAC provides component-specific robustness: conservative for safety-critical objectives and less conservative for auxiliary ones.

# D IMPLEMENTATION DETAILS

This appendix provides full details of the uncertainty model, network architectures, base algorithms, and hyperparameters used in our experiments.

## D.1 UNCERTAINTY MODEL

The uncertainty set $\mathcal{P}_{s,a}$ models potential variations in dynamics between training and deployment. Following Queeney & Benosman (2023), we implement $\mathcal{P}_{s,a}$ through controlled perturbations of nominal transitions. For each $(s, a)$, we sample a latent variable $x \sim \mathcal{U}([-\epsilon, \epsilon]^d)$ and perturb the nominal next state as

$$s' \mapsto s + (s' - s)(1 + x),$$

where $\epsilon$ controls the perturbation magnitude. This approximates variations such as friction, mass, or actuator response changes in robotics. The framework is general: domain-specific perturbations can replace this generic scheme if available. Importantly, MPAC requires only the ability to sample transitions (generative access), not adversarial training or multiple simulators.

## D.2 RISK-ADJUSTED VALUE ESTIMATION

Each risk measure $\rho$ is characterized by a distortion function $g : [0, 1] \to [0, 1]$ (Definition 3). For $n$ perturbed transitions from $\mathcal{P}_{s,a}$, we sort their Bellman targets and assign weights

$$w_\rho^{(i)} = g\left(\tfrac{i}{n}\right) - g\left(\tfrac{i-1}{n}\right). \tag{28}$$

The sample-based estimate for component $i$ is

$$\hat{Q}_i^\pi(s,a) = R_i(s,a) + \gamma \sum_{j=1}^n w_{\rho_i^+}^{(j)} Q_i(s_j', \pi(s_j')), \qquad (29)$$

with analogous estimation for the cost critic using $\rho^-$. This weighted averaging yields risk-adjusted estimates compatible with TD learning.

### D.3 NETWORK ARCHITECTURES

All critics and the policy network use three hidden layers with 256 units, ELU activations, and layer normalization followed by $\tanh$ after the first layer (Abdolmaleki et al., 2020). The policy outputs the mean and diagonal covariance of a Gaussian $\pi(a|s) = \mathcal{N}(\mu(s), \Sigma(s))$, with $\Sigma(s)$ stabilized via softplus. Each reward component critic $Q_{\theta_i}$ and the cost critic $Q_{\theta_c}$ have independent parameters but identical architectures. Target networks are updated with exponential moving average ($\tau = 0.005$).

### D.4 ALGORITHM CONFIGURATIONS

We use Maximum a Posteriori Policy Optimization (MPO) (Abdolmaleki et al., 2018) as our base policy optimization method. MPO operates through a two-stage process: first computing a non-parametric policy improvement step subject to a KL divergence constraint $\epsilon_{KL} = 0.1$, then projecting this improvement onto the parametric policy while respecting separate KL constraints for the mean ($\beta_\mu = 0.01$) and covariance ($\beta_\Sigma = 10^{-5}$) updates. The algorithm uses 20 action samples per update and maintains dual variables for the KL constraints, updated with a learning rate of 0.01.

For safety constraints, we integrate CRPO (Xu et al., 2021), which dynamically alternates between reward maximization and cost minimization based on batch-level constraint satisfaction. When the estimated constraint is satisfied, the algorithm optimizes for reward; otherwise, it prioritizes cost minimization. Following Liu et al. (2022), we compute exact temperature parameters at each update to ensure stable switching between objectives.

The MPAC-specific components use risk estimation with $n = 5$ perturbed samples per transition, where perturbations have magnitude $\epsilon = 0.1$ scaled by the empirical state standard deviation. The influence-based weight computation uses a temperature parameter $\tau = 0.1$. Component-specific risk measures are assigned based on objective criticality: safety-critical components use conservative CVaR with $\alpha = 0.1$, secondary objectives use $\alpha = 0.3$, and auxiliary objectives use $\alpha = 0.5$, as detailed in Appendix G.

## E EXPERIMENTAL SETUP

We evaluate all algorithms on continuous control tasks from the Real-World RL Suite (Dulac-Arnold et al., 2020). Each task includes predefined safety constraints and supports multi-objective reward decomposition. All methods use identical network architectures and training procedures to ensure fair comparison. We consider safety constraints with a budget of $B = 100$ for all tasks. Table 3 describes the specific constraints used, which reflect real-world operational limits critical in robotic systems.

To evaluate robustness, we test policies on perturbed environments where key physical parameters vary from training conditions:

For each algorithm we train on the nominal environment and evaluate across multiple seeds and perturbation levels, aggregating results over tasks using the interquartile mean (IQM) and performance profiles from the `rliable` toolkit (Agarwal et al., 2021). Confidence intervals are computed via stratified bootstrapping across seeds $\times$ perturbations.

## F ADDITIONAL RESULTS

This section provides further analyses complementing the main results. Figure 3 compares adaptive influence-based weights to fixed-weight schemes, showing that state-dependent adaptation con-

Table 2: Hyperparameter settings used in all experiments

| Parameter | Value |
|---|---|
| *Training* | |
| Batch size per update | 256 |
| Environment batch size | 3,000 |
| Initial environment batch size | 10,000 |
| Updates per environment step | 1 |
| Total training steps | 500,000 |
| Evaluation frequency | 10,000 steps |
| Checkpoint frequency | 50,000 steps |
| *Optimization* | |
| Actor learning rate | $3 \times 10^{-4}$ |
| Critic learning rate | $3 \times 10^{-4}$ |
| Discount factor ($\gamma$) | 0.99 |
| Target network EMA ($\tau$) | 0.005 |
| *MPO Parameters* | |
| Non-parametric KL ($\epsilon_{KL}$) | 0.1 |
| Action penalty KL | $10^{-3}$ |
| Mean KL ($\beta_\mu$) | 0.01 |
| Covariance KL ($\beta_\Sigma$) | $10^{-5}$ |
| Action samples | 20 |
| Dual learning rate | 0.01 |
| *Environment* | |
| Episode horizon | 1,000 steps |
| Evaluation trajectories | 3 (deterministic) |
| Safety budget ($B$) | 100 |

Table 3: Safety constraints for all tasks

| Task | Safety Constraint | Safety Coefficient |
|---|---|---|
| Cartpole Swingup | Slider Position | 0.30 |
| Walker Walk | Joint Velocity | 0.25 |
| Walker Run | Joint Velocity | 0.30 |
| Quadruped Walk | Joint Angle | 0.15 |
| Quadruped Run | Joint Angle | 0.30 |

sistently improves both reward and cost outcomes. Figure 4 evaluates robustness under increasing perturbation magnitudes, where MPAC degrades gracefully while maintaining advantages over baselines. Finally, Figure 5 presents reward and cost rankings across environments, confirming that MPAC consistently secures top ranks across tasks while balancing safety and performance. Together, these analyses highlight the complementary roles of dynamic weighting and component-specific robustness in enabling strong generalization across varied deployment conditions.

Taken together, these analyses reinforce the main findings: adaptive influence-based weighting is essential for balancing objectives across states, risk-aware value functions improve safety without sacrificing performance, and robustness is maintained under a broad range of perturbations. The ablations confirm that removing either dynamic weighting or component-specific risk significantly degrades performance, highlighting their complementary roles.

Table 4: Perturbation ranges for test environments

| Domain | Parameter | Nominal | Test Range |
|--------|-----------|---------|------------|
| Cartpole | Pole Length | 1.00 | [0.75, 1.25] |
| Walker | Torso Length | 0.30 | [0.10, 0.50] |
| Quadruped | Torso Density | 1,000 | [500, 1,500] |

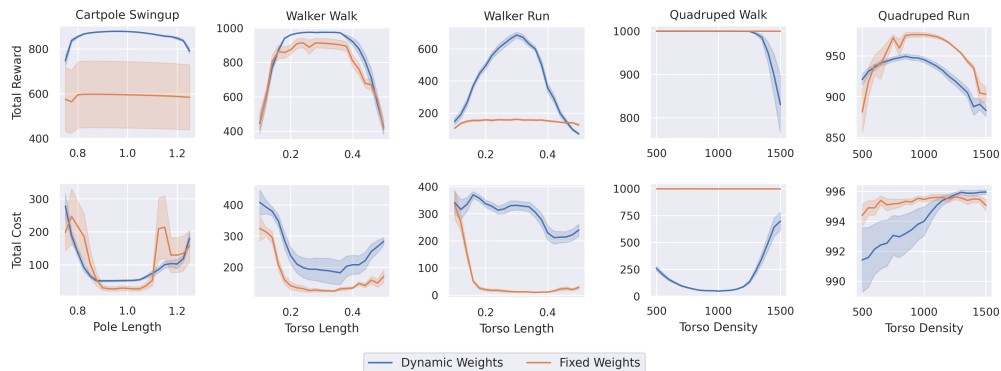

Figure 3: Dynamic vs. fixed weights. Adaptive influence-based weighting consistently improves both reward and cost performance compared to fixed weighting, demonstrating the benefits of state-dependent adaptation.

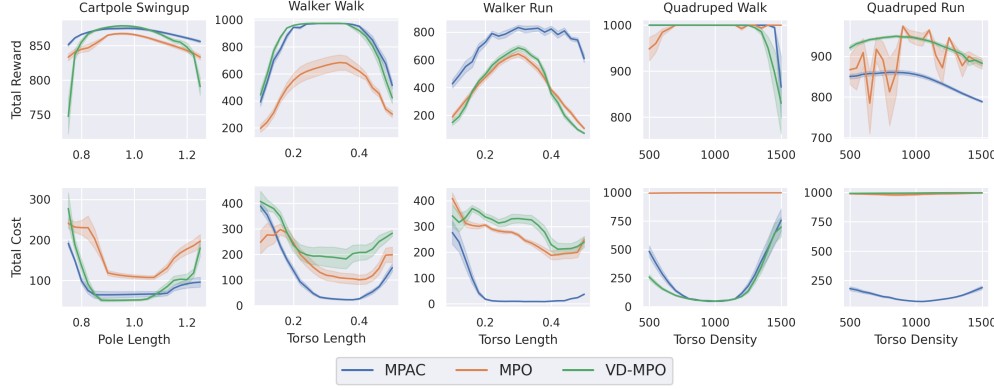

Figure 4: Robustness to perturbations. MPAC degrades gracefully as perturbation magnitude increases, maintaining higher reward and lower cost compared to all baselines. This confirms that component-specific robustness extends across a range of environment shifts.

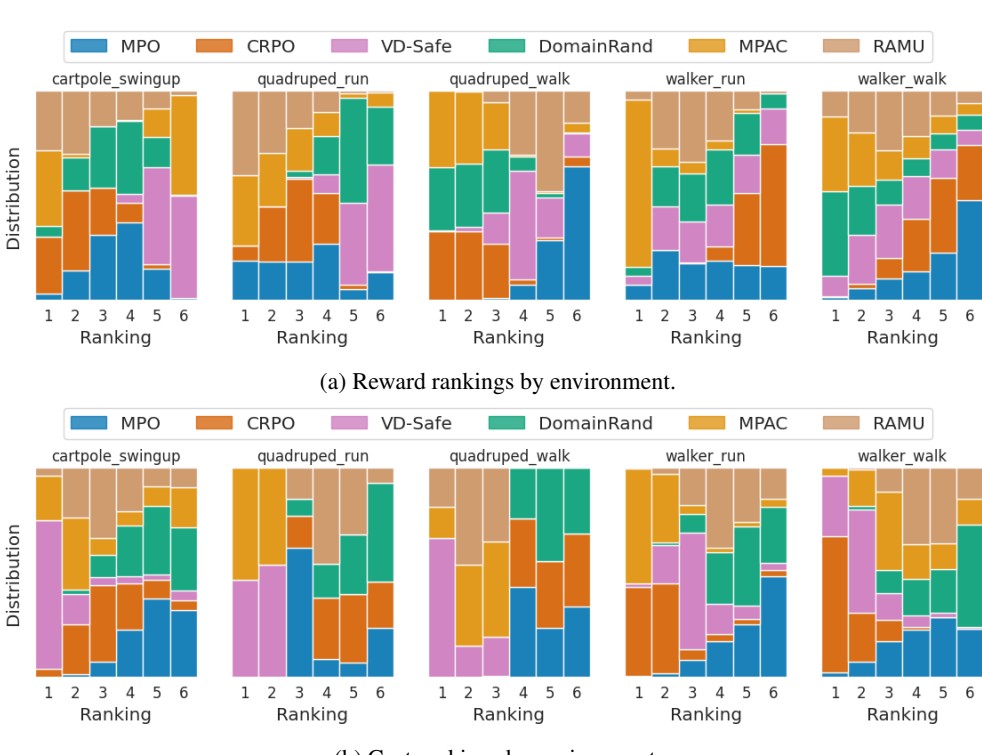

Figure 5: Ranking distributions across environments and perturbations. MPAC achieves top reward ranks while preserving strong safety performance, confirming its robustness across settings.

# G  REWARD COMPONENT DECOMPOSITION

This appendix details the reward decomposition used in MPAC experiments. Each environment's reward is decomposed into interpretable components that capture different aspects of task performance. Components are assigned risk measures (CVaR parameters) based on their criticality: primary objectives use more conservative evaluation (higher CVaR $\alpha$) while auxiliary objectives use less conservative measures.

## G.1  CARTPOLE SWINGUP

The Cartpole task requires balancing a pole upright while maintaining cart position. We decompose the reward into three components:

Table 5: Cartpole swingup reward components

| Component | Formula | CVaR | Priority |
|---|---|---|---|
| Pole stability with control | $\frac{\cos(\theta)+1}{2} \cdot \frac{4+\tau_{\text{ctrl}}}{5}$ | 0.5 | Primary |
| Smooth movement | $\frac{1+\tau_{\text{vel}}}{2} \cdot \frac{1+\tau_{\text{pos}}}{2}$ | 0.3 | Secondary |
| Base task reward | Original environment reward | 0.1 | Tertiary |

Where $\tau_{\text{ctrl}}$, $\tau_{\text{vel}}$, and $\tau_{\text{pos}}$ are tolerance functions that return values in $[0, 1]$ based on control effort, angular velocity, and cart position respectively. The primary component combines pole angle (most critical) with control efficiency. The secondary component encourages smooth, centered motion.

## G.2  WALKER WALK/RUN

The Walker tasks involve bipedal locomotion with different target speeds. The decomposition captures forward progress, posture, and stability:

Table 6: Walker reward components

| Component | Formula | CVaR | Priority |
|---|---|---|---|
| Forward progress | Base environment reward | 0.5 | Primary |
| Upright posture | $\frac{1+\text{torso\_upright}()}{2}$ | 0.3 | Secondary |
| Standing stability | $\frac{3 \cdot \tau_{\text{height}} + \tau_{\text{upright}}}{4}$ | 0.1 | Tertiary |

The primary objective focuses on achieving the target velocity. Maintaining upright posture is secondary, while overall standing stability (combining height and orientation) is treated as an auxiliary objective.

## G.3  QUADRUPED WALK/RUN

The Quadruped tasks require coordinated four-legged locomotion. Components balance speed, stability, and coordination:

Table 7: Quadruped reward components

| Component | Formula | CVaR | Priority |
|---|---|---|---|
| Forward velocity | $\tau_{\text{vel}}(v_x, v_{\text{target}}, \text{margin} = v_{\text{target}})$ | 0.5 | Primary |
| Upright orientation | $\tau_{\text{upright}}(\text{torso\_angle}, \text{margin} = 1)$ | 0.3 | Secondary |
| Coordinated movement | Forward velocity $\times$ Upright orientation | 0.1 | Tertiary |

The tolerance function $\tau_{\text{vel}}$ returns 0.5 when velocity equals zero and 1.0 when reaching the target speed. The combined reward encourages maintaining stability while moving.

## G.4 RISK ASSIGNMENT STRATEGY

The CVaR assignments follow a consistent pattern across environments:

- **CVaR 0.5** (least conservative): Primary task objectives critical for success
- **CVaR 0.3** (moderately conservative): Stability and form constraints
- **CVaR 0.1** (most conservative): Auxiliary objectives or combined rewards

This inverted assignment (higher $\alpha$ for critical components) reflects that in our implementation, higher CVaR values focus on better outcomes rather than worse outcomes. The strategy ensures that primary objectives are achieved reliably while allowing more variability in auxiliary objectives.

All components are normalized to $[0, 1]$ using tolerance functions from the DMC suite (Tunyasuvunakool et al., 2020), ensuring consistent scaling across objectives. The decomposition enables MPAC to apply appropriate risk attitudes to different aspects of complex tasks, achieving robust performance without excessive conservatism.

