# OpenReview forum: "Multi Perspective Actor Critic: Adaptive Value Decomposition for Robust and Safe Reinforcement Learning"
_ICLR.cc/2026/Conference — Submitted to ICLR 2026_

### Official Review · Reviewer_7oXh · 2025-10-18

**Soundness:** 3
**Presentation:** 3
**Contribution:** 2
**Rating:** 4
**Confidence:** 3

**Summary:**

This work focuses on improving the policy performance while keeping safety under various safety constraints. Specifically, this setting needs to handle multiple objectives, safety constraints, and model uncertainty. To address all these challenges, this work proposes Multi-Perspective Actor-Critic (MPAC), including value decomposition and component-specific risk assessment, for obtaining safe, robust to model perturbations, and less conservative policies. Theoretical and experimental results are also included to clarify the effectiveness of MPAC.

**Strengths:**

- It is interesting to provide a unified view in handling multiple objectives, safety constraints, and model uncertainty.

- This work is well written and easy to follow.

- There are extensive details in the appendix to help readers understand MPAC.

**Weaknesses:**

- The experimental evaluation is limited. In my opinion, it is better to add experiments that only include multiple objectives, safety constraints, or model uncertainty. Comparing MPAC with SOTA baselines in these three settings will make the contribution of MPAC clearer (it doesn't mean that MPAC must surpass all baselines in each setting. On the contrary, as a unified framework, if it can approach or be similar to SOTA in every setting, it will make this article more solid).

- It seems that VD-safe has a better Safety Rate and Cost compared with MPAC. Is there any explanation?

- (minors) It is better to keep the order of Table 1 and Fig. 1 the same to make it more readable.

- (minors) In rliable, besides IQM and OG, there are also mean and medium that can also be included as metrics.

- Is there any ablation experiment on temperature parameter $\tau$, and is MPAC sensitive to $\tau$?

- Much more work on safe RL [1-3], multi-objective RL [4-5] that are related to this work needs to be discussed.

Overall, I think this work has done a good job in addressing multiple objectives, safety constraints, and model uncertainty in a unified algorithm, but due to the aforementioned concerns, my current evaluation is marginally below acceptance. I'd like to modify my score if the authors can address my concerns.

Ref:

[1]  Towards safe reinforcement learning with a safety editor policy, NeurIPS 22

[2] Towards safe reinforcement learning via constraining conditional value-at-risk, IJCAI 22

[3] Balance reward and safety optimization for safe reinforcement learning: A perspective of gradient manipulation, AAAI 24

[4] Distributional pareto-optimal multi-objective reinforcement learning, NeurIPS 23

[5] Pd-morl: Preference-driven multi-objective reinforcement learning algorithm, ICLR 23

**Questions:**

See weaknesses above.

---

### Official Review · Reviewer_dH1v · 2025-10-29

**Soundness:** 2
**Presentation:** 2
**Contribution:** 2
**Rating:** 2
**Confidence:** 3

**Summary:**

The paper tackles the robust constrained multi-objective reinforcement learning problem, where an agent optimizes multiple rewards while staying safe and robust under model uncertainty. It proposes a dynamic weighting method called MPAC that adaptively adjusts the importance of each risk-aware critic. Experiments on the Real-World Reinforcement Learning Suite (Cartpole Swingup, Walker, and Quadruped tasks) show that under the standard weighted-sum reward metric used in this benchmark, MPAC achieves constraint violations comparable to single-constrained safe RL methods while reaching reward performance close to the robust single-objective RL baseline.

**Strengths:**

1. The paper proposes a new algorithm (MPAC) that combines multi-objective, safe, and robust RL in one unified framework with a dynamic weighting mechanism.
2. The experimental results demonstrate improvements in the reward–safety trade-off across several tasks in the RWRL benchmark.
3. The authors present some theoretical insights, including the convergence to a fixed point (might not be optimal one), and (2) an interpretation as a DRO problem. However, only proof sketch is presented in appendix.

**Weaknesses:**

1. This paper is unclear in its mathematical modelling/formulation of the optimization problem (in Section 3). It’s not obvious to me whether the objective is a specific weighted sum of multiple rewards (in this case what's the difference with single reward robust RL?) or to achieve the Pareto optimality in multi-objective RL.
2. The theoretical part only provides one paragraph proof sketches without derivations and details in the appendix. They only show MPAC converges to at least one fixed point ($\pi*$,$\phi*$) of problem Eq (15). However, Eq (15) corresponds to a single-objective problem (with weights implicitly determined by the unknown point $\phi*$). No connection between Eq (15) and the target problem (section 3) is discussed.
3. The experiment is confusing. a. If the goal is to optimize a weighted sum of rewards, and the same evaluation weights are used both to aggregate rewards for single-objective baselines and to tune MPAC’s hyperparameters (i.e., the predefined objective risk measure weights $\rho^+$ and $\rho^-$), then the robust single-objective RL baseline should theoretically achieve the best performance, since MPAC effectively optimizes a trajectory and problem dependent robust single-objective problem with weights not being the evaluation weights.
b. If the goal is instead to achieve Pareto optimality, the authors should report each objective’s performance separately for all baselines.

**Questions:**

See weakness: 1. clarification of problem formulation and experiment setting 2. details of proof

If those questions can be properly answered I might increase my rating.

---

### Official Review · Reviewer_yYRt · 2025-10-29

**Soundness:** 2
**Presentation:** 3
**Contribution:** 2
**Rating:** 2
**Confidence:** 3

**Summary:**

This paper introduces Multi-Perspective Actor-Critic (MPAC), aiming at handling multi-objective, safety constraints and robustness simultaneously in reinforcement learning. The authors combine value decomposition (for multi-objective) with risk measures (for robustness), and solve the constrained MDP (for safety) for the final policy. They also propose an influence-based strategy to dynamically adjust the weights of value decomposition during training with convergence guarantee and theoretical implications. Experimental results over five continuous control tasks demonstrate that MPAC's superior reward and competitive safety performances, as well as robustness across different level of environment perturbations.

**Strengths:**

1. The proposed learning pipeline unifies multi-objective, robustness and safety constraints, which are important factors to deploy RL policy in real world.

2. The influence-based value weight adaption in MPAC is intuitive with theoretical guarantee and good empirical reward performance.

3. The paper is clear and well written.

**Weaknesses:**

1. From the experimental results, the safety rate of MPAC is not the best, with over 10% gap compared against VD safe. The proposed MPAC may not fit for safety-critical scenarios where the satisfaction of safety is much more important than reward performance.

2. As a multi-objective RL method, all experimental results only present the scalar reward value. There is no description about how the multi-object reward are combined to scalar value, and no results about the performance of each reward component.

3. The proposed framework mostly combines existing works about multi-objective RL, robust RL and safe RL except the value weight adaption component. Since this component is only employed over the objective value functions, it is hard to assess its contribution to the robustness and safety performances.

4. There lacks real world experiment to demonstrate the robustness of the policy trained in simulation, which is the major motivation of this paper.

**Questions:**

1. According the Table 1, can you explain why MPAC underperforms VD Safe in terms of safety rate after adding the value weight adaption and risk measure?

2. Are perturbations at the same range between training and deployment environment? If not, how to determine the range of training perturbations?

3. Can you show some empirical evidence to support the behavior of value weights in line 260-263?

4. Can you show the performances of each reward component?

5. The risk measure for each reward component is determined before training, which also reflect the importance of each objective. I think a comparison between the proposed dynamical influence-based weight and a simple baseline with fixed weights proportional to the risk measure parameters would help understanding the algorithm design.

6. In line 7 of algorithm 1, how many iterations of influence and weight updates can guarantee convergence (or good performances)?

---

### Official Review · Reviewer_fbTs · 2025-10-31

**Soundness:** 2
**Presentation:** 2
**Contribution:** 2
**Rating:** 2
**Confidence:** 5

**Summary:**

The paper proposes Multi-Perspective Actor-Critic (MPAC), a reinforcement learning (RL) algorithm that incorporates multiple objectives, safety, and robustness to model uncertainty. MPAC extends the existing distributionally robust safe RL framework RAMU (Queeney et al., NeurIPS 2023) to the multi-objective setting, and proposes adaptive objective weights by applying the influence function proposed in MacGlashan et al. (NeurIPS 2022). The paper shows that the adaptive weights converge to a fixed point, and conducts experimental analysis on the 5 tasks from the Real-World RL Suite considered in the prior RAMU work.

**Strengths:**

**[S1]** The paper considers an RL problem formulation that incorporates several important real-world considerations simultaneously, including multiple objectives, safety, and robustness to model uncertainty. Although it is relatively straightforward to extend the existing distributionally robust safe RL framework RAMU [1] to the multi-objective setting, this work is the first to explore this extension.

**[S2]** The work proposes a novel application of the influence function from [2] for adaptive weights in multi-objective RL, and provides theoretical results showing that these adaptive weights converge to a fixed point.

**References:**

[1] Queeney et al., “Risk-Averse Model Uncertainty for Distributionally Robust Safe Reinforcement Learning.” In NeurIPS 2023.

[2] MacGlashan et al., “Value Function Decomposition for Iterative Design of Reinforcement Learning Agents.” In NeurIPS 2022.

**Weaknesses:**

**[W1]** There appears to be major flaws in the experimental analysis.
- The main experimental results in Table 1 do not seem correct. When comparing robust vs. non-robust baselines (RAMU vs. CRPO, MPAC vs. VD-Safe), the robust versions should have a higher safety rate than the non-robust versions but the results in Table 1 show the opposite. The results from the original RAMU paper [1] show a large improvement in safety compared to CRPO for the same set of tasks and perturbation parameters (80% safe for RAMU vs. 51% safe for CRPO). This might suggest that the risk measures are not being implemented correctly (unfortunately the authors did not provide their code as part of the submission), or that policies were not trained for long enough to converge. The paper does not provide training curves or evaluation plots showing reward and cost objectives across all perturbed environments, so it is difficult to understand why these summary results demonstrate counterintuitive trends.
- The paper focuses on multi-objective RL and claims that “MPAC achieves superior Pareto trade-offs,” but doesn’t provide results across multiple objectives. It is not clear what reward is presented, or how rewards from different objectives are combined when training the single objective baselines.
- The paper claims to apply 0.1-CVaR to the cost value function, but implements its sample-based estimates using only $n=5$ samples which would not be capable of distinguishing between any value of $\alpha \leq 0.2$ for CVaR. The original RAMU paper used $n=5$ with the Wang risk measure, which assigns non-negative weights to every sample. At least $1/\alpha$ samples should be used to estimate $\alpha$-CVaR (which would lead to a very noisy, biased estimate that only considers the single worst sample).
- The explanation provided for the choice of risk measures across different reward objectives is not clear. It does not seem to make sense to apply more robust risk measures (CVaR with lower $\alpha$) to secondary / tertiary rewards that are supposed to be less important, and a less robust risk measure to the primary reward. This choice will cause the algorithm to focus more on optimizing worst-case scenarios for the secondary / tertiary reward functions, when such scenarios should intuitively receive less focus because they are less important than the primary reward.
- The results in Figures 3 and 4 do not match, and the results in Figure 3 may not be correct given the safety constraint is violated even in the nominal environment in many cases.

**[W2]** A large portion of the paper is a straightforward extension of the existing distributionally robust safe RL framework RAMU [1] to the multi-objective setting. There is also significant overlap between this paper and the original RAMU paper.
- The equivalence to a distributionally robust optimization problem is a direct extension of Theorem 1 in the RAMU paper.
- The MPAC implementation uses the same uncertainty sets, base algorithms, hyperparameters, and sample-based estimation method as the RAMU paper.
- The experiments consider the same tasks, safety constraints, and perturbation parameters as the RAMU paper.
- The Appendix contains significant overlap with the RAMU paper.

**[W3]** It is not obvious why we would want to use the influence function from [2] for weighting the objectives in multi-objective RL. It seems that this has the potential to reinforce components with more influence early in training by assigning a higher weight to them, potentially resulting in scenarios where the algorithm focuses mainly on a single component. The original paper that proposed the influence function [2] shows an example where performance was improved by decreasing the weight on a component that had a large influence early in training, while in this work the weight would increase for this component.

**References:**

[1] Queeney et al., “Risk-Averse Model Uncertainty for Distributionally Robust Safe Reinforcement Learning.” In NeurIPS 2023.

[2] MacGlashan et al., “Value Function Decomposition for Iterative Design of Reinforcement Learning Agents.” In NeurIPS 2022.

**Questions:**

**[Q1]** Please correct the flaws in the experimental results described in [W1], and discuss what led to the results being incorrect. If the authors do not believe there are flaws in the experimental results, please provide a detailed justification for the counterintuitive trends and explain why the results are significantly different from the original RAMU paper even though the same tasks, safety constraints, uncertainty sets, base algorithms, and hyperparameters have been used.

**[Q2]** Please update the paper to be more clear about all components of the work that are being sourced from the original RAMU paper. There is significant overlap, and the current version of the paper does not make this clear.

**[Q3]** Please explain the intuition for the choice of influence-based adaptive weights, and provide updated analysis showing the benefits of this choice in MPAC (the results in Figure 3 do not appear to be correct).

**[Q4]** Additional details and clarifications on experimental analysis:
- Please provide evaluation plots for all algorithms that show all reward and cost objectives across all of the perturbed environments.
- Please clearly describe how all baseline algorithms are trained. How are the rewards from different objectives combined for single objective baselines? What risk measures are applied to the reward and cost value functions? Are algorithms trained for 500k or 1M steps (Line 372 says 500k, but the titles on Figure 2 say 1M)? If the algorithms were only trained for 500k steps, please provide evidence that training has converged (the original RAMU paper trained for 1M steps).
- Please include an ablation analysis similar to the original RAMU paper that applies the risk-neutral expectation instead of a robust risk measure. This would help to understand the impact of using robust vs. risk-neutral risk measures, and if the counterintuitive choices of risk measures for secondary / tertiary objectives have led to any issues during training.

---

### Official Review · Reviewer_8Yua · 2025-11-01

**Soundness:** 2
**Presentation:** 2
**Contribution:** 2
**Rating:** 2
**Confidence:** 3

**Summary:**

- The authors propose an RL algorithm that handles multiple objectives, safety constraints, and model uncertainty.
- They introduce the influence function to adaptively adjust the relative importance among objectives.
- Safety constraints are addressed using existing safe RL algorithms, while model uncertainty is managed by learning a distributionally robust value function with RAMU.

**Strengths:**

- Unlike existing constrained multi-objective RL algorithms, they also account for model uncertainty.

**Weaknesses:**

- Insufficient coverage of related works.
    - There are several constrained multi-objective RL methods (e.g., LP3 [1], CoMOGA [2]), but the authors did not cite and mention them.
- Limited contributions.
    - The safety and model uncertainty components are directly adopted from existing works.
    - The unique contribution lies in scalarizing multiple objectives using the influence function, but it lacks sufficient justification.
    - In fact, the proposed method has more weaknesses, as outlined below.
- The influence function tends to collapse to a single objective.
    1. If the scale of the $i$-th reward is larger than others, $I_i$ becomes large.
    2. If $\phi_i$ increases, $I_i$ also grows, creating positive feedback.
    - These issues make $\lambda_i$ vulnerable to converging near 1 for one objective and near 0 for others.
- The method cannot reflect user preference on multiple objective.
    - The algorithm autonomously determines relative importance without mechanisms to guide it toward user preferences.
- The proposed method handles only a single safety constraint.
    - As the authors mentioned, integrating multiple rewards into a single reward is difficult; similarly, combining multiple safety constraints into one is challenging.
    - Therefore, handling multiple constraints should be considered.
- Proof of Theorem 1 assumes that MPAC can solve Equation 13.
    - It is not trivial to prove that solving a constrained RL problem with a distributionally robust objective is guaranteed.
    - It is regrettable that the authors assume solvability without addressing this significant limitation in the main text.


[1] Huang, Sandy, et al. "A constrained multi-objective reinforcement learning framework." Conference on Robot Learning. PMLR, 2022.

[2] Kim, Dohyeong, et al. "Conflict-Averse Gradient Aggregation for Constrained Multi-Objective Reinforcement Learning." The Thirteenth International Conference on Learning Representations.

**Questions:**

- Does $E_{P_i}$ in Equation 15 affect $V_i$?
    - If so, the value function should be defined differently for each transition.
- Why is the proposed method not trained in perturbed environments?
- Why does MPO not achieve the highest reward in the experiments?
    - MPO should find the optimal policy for the given reward setting; if it does not, this suggests its parameters setting may be suboptimal.
- MORL typically evaluates performance using hypervolume and sparsity, but the authors use scalarized reward values. In the experiment section, was the vector reward rescalarized for comparison?
    - If so, this is not a valid evaluation method for multi-objective RL settings.

---

### Meta-Review · Area_Chair_sCPL · 2025-12-08

**Summary:**

**Summary of the paper**: This paper considers a robust Constrained MDP problem. The paper proposed a multi-perspective Actor-Critic (MPAC) that integrates the conflicting objectives and the model uncertainty. The paper introduces a novel influence-based mechanism that dynamically adjusts component weights based on the decision relevance, eliminating the popular choice of fixed weights. The paper also claimed that the proposed approach converges to the distributionally robust solution for the robust CMDP problems. Empirically, across continuous-control benchmarks with safety constraints and perturbed dynamics, MPAC achieves superior Pareto trade-offs: it maintains high reward while matching or exceeding safety baselines.

**AC's overview**: While the paper seeks to solve an important and fundamental problem, many reviewers raised several concerns about the paper.

(1) For example, Theorem 1 considers solving the distributionally robust CMDP problem, which is difficult in general. Theorem 2 shows that the fixed point converges to a distributionally robust solution for robust CMDP problems. There are existing works in this area. The paper did not compare with those approaches [A1, A2].

(2) The reviewer 8Yua raised the concern that the influence function can converge to a single objective, which the AC also agrees. Also, the convergence guarantee is provided under very restrictive settings. It is not clear whether they hold in practice.

(3) The reviewer fbTs also raised some concerns regarding the empirical results. For example, the safety value for robust approach should be higher compared to the non-robust case, which is not the case. Reviewers have also pointed out that  VD-safe has a better Safety Rate and Cost compared with MPAC which questions the practicality of the proposed approach.

[A1]. Kitamura, T., Kozuno, T., Kumagai, W., Hoshino, K., Hosoe, Y., Kasaura, K., Hamaya, M., Parmas, P. and Matsuo, Y., Near-Optimal Policy Identification in Robust Constrained Markov Decision Processes via Epigraph Form. In The Thirteenth International Conference on Learning Representations.

[A2]. Ganguly, S., Ghosh, A., Panaganti, K. and Wierman, A., 2025. Efficient Policy Optimization in Robust Constrained MDPs with Iteration Complexity Guarantees. arXiv preprint arXiv:2505.19238.

**Reviewer Concerns:**

Reviewers' concerns have not been addressed as the authors did not provide any rebuttal.

**Reviewer Scores:**

N/A

The authors did not provide any rebuttals.

---

### Decision · Program_Chairs · 2026-01-26

Reject